# Large losses from little lies: Strategic gender misrepresentation and cooperation

**Michalis Drouvelis**[1,2] , **Jennifer Gerson**[3] , **Nattavudh Powdthavee**[4,5] *, **Yohanes E. Riyanto**[5]

**1** Department of Economics, University of Birmingham, Birmingham, United Kingdom, **2** Center for Economic Studies, University of Munich, Munich, Germany, **3** School of Health and Psychological Sciences, University of London, London, United Kingdom, **4** Warwick Business School, University of Warwick, Coventry, United Kingdom, **5** Division of Economics, Nanyang Technological University, Singapore, Singapore

☺ These authors contributed equally to this work.
* nick.powdthavee@ntu.edu.sg

**Data Availability Statement:** All data and STATA .do files are available from the following URL: https://github.com/npowdthavee/largelossesfromlittlelies.

## Abstract

This paper investigates the possibility that a small deceptive act of misrepresenting one's gender to others reduces cooperation in the Golden Balls game, a variant of a prisoner's dilemma game. Compared to treatments where either participants' true genders are revealed to each other in a pair or no information on gender is given, the treatment effects of randomly selecting people to be allowed to misrepresent their gender on defection are positive, sizeable, and statistically significant. Allowing people to misrepresent their gender reduces the average cooperation rate by approximately 10–12 percentage points. While one explanation for the significant treatment effects is that participants who chose to misrepresent their gender in the treatment where they were allowed to do so defect substantially more, the potential of being matched with someone who could be misrepresenting their gender also caused people to defect more than usual as well. On average, individuals who chose to misrepresent their gender are around 32 percentage points more likely to defect than those in the blind and true gender treatments. Further analysis reveals that a large part of the effect is driven by women who misrepresented in same-sex pairs and men who misrepresented in mixed-sex pairs. We conclude that even small short-term opportunities to misrepresent one's gender can potentially be extremely harmful to later human cooperation.

## Introduction

Gender differences in individual's willingness to cooperate represent one of the core research areas across many disciplines, including economics [1–3], psychology [4, 5], and neuroscience [6]. While it is commonly believed that women are more cooperative than men, there is conflicting evidence for this assertion. Small differences in contexts and experimental designs have produced results where women are more cooperative than men in some experiments and less cooperative than men in others—see [6, 7] for a comprehensive review of the literature. Yet, despite the mixed results, women are stereotypically expected to be more altruistic than men in almost all studies where the participants' gender is made salient. For example, in

**Funding:** The author(s) received no specific funding for this work.

**Competing interests:** The authors have declared that no competing interests exist.

experiments where the gender of the co-participant(s) is made available to the individual, men and women tend to expect the other women in their group to make choices that are kinder to external parties in coordination games [8], as well as to cooperate substantially more than male co-participants in prisoner's dilemma games [9].

To the best of our knowledge, previous findings on gender differences in cooperation are based on experiments where the participants either know or are not aware of their co-participants' gender. As a result, the existing studies are silent on the potential implications of allowing people to misrepresent their gender to others in a social dilemma interaction. Although the issue around gender misrepresentation may not be relevant in the past, modern technology—notably, the invention of social media—has undoubtedly made anonymized behaviour online a prevalent part of many people's daily interactions. This raises an important question: Might this new ability to hide one's gender be corrosive to human cooperation? Do 'bad apples' always lie about their gender identities to win others' trust for financial gain at their expense later? Or do they only act uncooperatively because there are opportunities for them to strategically misrepresent their identity to others, thus making their uncooperative behaviour either easier to justify or stands a better chance of succeeding? Currently, there is little insight into this question, and the extent of social preferences when there is a real possibility of gender misrepresentation remains imperfectly understood.

This paper proposes a new empirical test of how gender misrepresentation might impact human cooperation. Through a series of randomised lab and online experiments, we test whether randomly selecting people who can misrepresent their gender negatively affects their willingness to cooperate in a variant of the one-shot prisoner's dilemma game. We find evidence that the average defection level (with real money stakes for the entire group) is roughly 10–12 percentage points higher compared to treatments where participants' true gender is either revealed or not mentioned in the experiment.

Evidence of the conversion rate from misrepresentation to defection is much stronger; conditioning on misrepresentation, the defection rates among women and men are approximately 32 percentage points, on average. We also find some suggestive evidence of strategic gender misrepresentation; conditioning on misrepresentation, the highest defection rates are found among women in same-sex pairs and men in mixed-sex pairs. One possible explanation for this is that because gender stereotypes are much more salient in mixed-sex pairs [5], women might believe that other women will be more cooperative when they play the game against other men, so playing as a male should increase the chance of a successful defection for women in a same-sex pair. On the other hand, men might hold a belief that women will be more cooperative with other women than with other men, so playing as a female should increase the chance of a successful defection for men in a mixed-sex pair. We finish by discussing the potential implications of our findings for social media information system providers and users.

It should be noted here that our paper is somewhat similar to [10], in which the paper discusses a game-theoretic model of strategic misrepresentation of information in a social dilemma interaction. In the model, this misrepresentation is done through pre-play communication. The information that is being misrepresented is the planned or intended action, e.g., whether to defect or to cooperate in a prisoner's dilemma game against their opponents before all players choose their actual actions simultaneously. In our experiment, it is not the intended action that is being misrepresented. Instead, it is about the personal identity, e.g., gender, of players. This misrepresented information changes the opponents' belief about the player's likelihood of choosing a softer and more cooperative stance in the game. Also, as an anecdotal example of this strategic and intentional gender misrepresentation, there is a case of an Alberta man who changed his gender identity in his government-issued birth certificate and driver's

license. He did this because of the knowledge that, as a man, his motor insurance costs would be far higher than it would be if he was a woman.

## Background and hypotheses

There is extensive literature devoted to understanding why people cooperate much more than predicted by standard economic models, which assume rational and self-interested behaviours (see [11, 12] for extensive reviews). One of the leading theories on cooperation is the theory of indirect reciprocity, which explains people's preferences for cooperation as a result of wanting to build trust and reputation that are essential in long-term interactions [13–15]. People may also cooperate because they fear altruistic punishment when individuals in a group incur a cost to punish free-riders for not pulling their weight in cooperative endeavours [16–19]. Other studies have found evidence that individuals' preferences for cooperation may have stemmed from early in life [20] or are driven by personal characteristics that are close to being fixed over time [4, 21].

One of the most studied questions in this area is whether there are substantial gender differences in social preferences. In an extensive review of the literature, Croson and Gneezy [7] find that the results of social dilemma games are somewhat mixed. Depending on the experimental design, women appear more cooperative than men in some experiments and less cooperative than men in others. For example, Frank et al. [22] demonstrate that the overall cooperation rates in a prisoner's dilemma game for females are substantially higher than for males. Ortmann and Tichy [1] find that women cooperate more than men in the first round of a repeated prisoner's dilemma game before converging to cooperate at the same rate as men in later rounds. By contrast, studies like [23, 24] find that men contribute substantially more than women in a public goods game setting. Some studies also find little differences in the cooperation rates for males and females [25, 26].

What explains why gender differences in cooperation are found in some studies and not others? According to Croson and Gneezy [7], one possible explanation for this is that gender differences in cooperation are context-dependent, and one such context is the gender composition of the group. For example, Balliet et al. [5] propose—and later demonstrate in a meta-analysis—that women are more cooperative than men in mixed-sex social dilemmas. They propose that this is because, in gender stereotypes (i.e., from evolutionary and sociocultural perspectives), women are often perceived as kinder, more caring, and less selfish than men [27, 28]. These stereotypes are more likely to be activated and salient in mixed-sex than same-sex interactions. The authors also show that men are more likely to defect in mixed-sex than same-sex interactions, as gender stereotypes about men are more salient in the former than the latter context.

In another study, Charness and Rustichini [3] find that men cooperate substantially less often in the same-sex session when observed by other in-group men, whilst women cooperate substantially more often in the same-sex session when observed by other in-group women. One explanation for their findings is that men want to signal to other men in their group that they are tough, whereas women want to signal to other women in their group that they are kind. Vugt et al. [4] show that men contribute more to their in-group members in a public goods game if their group competes against another group than if there is no intergroup competition. By contrast, women's contribution is relatively unaffected by intergroup competition. Their findings strongly suggest that men care more than women about protecting the welfare of their in-group members.

More recently, Cigarini et al. [9] show that men in same-sex pairs hold the lowest expectations about their partner's cooperation rates after making eye contact with each other. They

also have the lowest cooperation rates of all gender pairings. Women in mixed-sex pairs display the most positive belief about their partner's cooperation level, which suggests that women generally expect that men will be reciprocal to their expected kindness in a social dilemma interaction. Evidence on women's expectation in mixed-sex pairs appears to be corroborated by the evidence in men's contribution levels: overall, men are significantly more cooperative when they interact with a woman.

Taken together, what previous studies seem to suggest is that gender differences in cooperation are more likely to emerge in contexts where these gender stereotypes are activated and made salient to the participants. Men generally like to be seen as tough and unyielding, whereas women like to be seen as kind and cooperative. Although women want to signal to other in-group women that they are generally cooperative, there is also some evidence that women tend to believe that, compared to other women, men will be kinder to them in a social dilemma interaction. However, participants in previous experiments have no reason to doubt the gender of the other participants in a group or a pair. What happens to the cooperation rates if we randomly allow half of the participants to misrepresent their gender to the other half? What can previous studies on gender differences in cooperation tell us about the possible implications of gender misrepresentation in social dilemma games?

Based on Cigarini et al.'s [9] findings, one possibility might be that men who are randomly allowed to misrepresent their gender may want to strategically misrepresent themselves as females in same-sex pairs to activate the other male's expectation that women are cooperative. Similarly, women may also want to strategically misrepresent themselves as males in same-sex pairs to activate the other female's expectation that men will be nicer to them than other women. Men may also want to strategically misrepresent themselves as females in mixed-sex pairs if they believe that women strongly prefer to signal to the other woman that they are kind and, therefore, will cooperate more in same-sex pairs [3]. However, there may be less incentive for women in mixed-sex pairs to strategically misrepresent their gender to their partner, considering that men in same-sex pairs have the most negative expectation about each other's behaviour in a social dilemma interaction [9]. Overall, we can see that people may choose to strategically misrepresent their gender if, by doing so, they can increase the probability that the other participant will cooperate. Given that the payoff for defection when the other cooperates is bigger in social dilemma games, one hypothesis is that people who decide to misrepresent their gender will also go on to defect rather than choose to cooperate. In other words, people who choose to misrepresent their gender do so to maximise the probability of a successful defection, i.e., defecting when the other is cooperating.

What about the cooperation rates of the participants who are not allowed to misrepresent their gender? Provided that they are not aware of the possibility that their co-participants can strategically misrepresent their gender to them, then their expected cooperation rates would be the same as in a typical social dilemma experiment. However, they will likely defect irrespective of the gender pairing if they believe that their co-participant might be strategically misrepresenting their gender. Therefore, they are likely to go on to defect. This hypothesis would be consistent with studies that find uncertainty about the opponent's strategy and/or the possibility of increased lying during communication reduces the frequency of cooperation [29, 30].

Taken together, we can form our key hypotheses. First, the average cooperation levels will likely be lower in treatments where gender misrepresentation among participants is possible. How much lower may depend on several factors, including the nature of the social dilemma game, the compliance rates, i.e., the number of people who choose to misrepresent when given a chance, and their beliefs about the other participant's cooperation rates after gender misrepresentation. Second, in a condition where players are given random opportunities to misrepresent their gender identity, then theories predict that players who choose to misrepresent

should be more likely to follow the weakly dominant strategy and steal since people will only choose to misrepresent if they perceive that by doing so, they can maximise the probability of a successful defection. Third, in a condition where players who are randomly forced to misrepresent their gender (and thus did not choose to) should not be more likely to steal than players who are not misrepresenting. And finally, in treatments where one-half of the sample is not allowed to misrepresent, the participants who are playing against someone who might be lying about their gender should also be more likely to steal to prevent their co-participant from winning all the money through defection.

## Materials and methods

### Participants

We recruited a total of 966 subjects– 686 students and 280 online participants on Prolific (www.prolific.co)–to participate in a one-shot variant of the prisoner's dilemma experiment with real money stakes. We ran the experiment on the student sample in Warwick Business School's laboratory in June 2019 and January 2020 and at Nanyang Technological University (NTU) in Singapore in August 2019 and January 2020. We then carried out the same experiment with the same treatments online on the Prolific sample in March and April 2020. We recruited a near-gender-balanced sample of students for the lab experiment: 50.8% and 51.1% were male participants in Warwick and NTU, respectively. However, it was not possible to do the same for the Prolific sample, ending up with 35.4% male participants for the online experiment. All participants in both lab and online experiments gave written consent. Since we can only ethically use adult sample, 2 participants under 18 were dropped from the final sample after reviewing the data, hence we included no minor in the study.

### Experimental details

The University of Warwick Research Governance and Ethics Committee reviewed and approved this research. All authors confirm that the experiment was performed following the University of Warwick Research Governance and Ethics Committee's regulations.

In our experiment, which we pre-registered the research plan and hypotheses through the Open Science Framework (O.S.F.; https://osf.io/5q4hv), we randomised participants into the following four treatments:

1. Blind

2. True gender

3. Randomly assigned opportunity to misrepresent gender

4. Randomly assigned gender

In two of these treatments—treatments 3 and 4—we randomly assigned one participant in each pair either i) an opportunity to misrepresent their gender to the other participant or ii) a gender that is either the same or different from their true gender. We also informed the other person in a pair that their partner was given an opportunity to misrepresent their gender (Treatment 3) or was randomly assigned a gender that is either the same or different from their true gender (Treatment 4). The purpose of Treatment 4 is to enable us to test whether i) exogenously forcing one partner to have a different gender from their true gender and ii) giving the remaining partner the idea that the other player's reported gender may or may not be the same as their true gender corrode cooperation similarly as if the decision to misrepresent is endogenously determined.

We ran each treatment in random sessions. In our lab experiments, participants were students from various majors, and they were randomly assigned to one of the four main treatments. Once participants had arrived at the lab, they were randomly assigned to different numbered seats with partitions. The lab experiments were implemented using the z-Tree program [31]. At the beginning of the experiment, we read the experimental instructions out loud to participants.

Likewise, the experimental protocol for online experiments was largely the same. We randomly assigned participants to one of the four main treatments. The online experiments were implemented using the o-Tree program (https://www.otree.org). Participants were redirected via a link from Prolific to the server where we hosted our o-Tree program.

At the end of the experiment, all participants had to complete a questionnaire about their socio-demographic status, including age, gender, and—for the student sample—what subject they were majoring in at the university. After that, they received their payment.

In all treatments, two randomly matched players played a game called the "Golden Balls" game, which we adopted from a popular game show on T.V. in the U.K. and the Netherlands [32, 33]. Each player had to decide whether to 'split' or 'steal' the money in a pot. If both players cooperated to split, they each received £10. If one chose to steal and the other chose to split, the person who stole received £20, and the person who cooperated received nothing. However, if both players stole, then both received nothing. Fig 1 displays the payoff matrix of the Golden Balls game that we showed the participants. Note that with minimum wages of £7.70 per hour in the U.K. for those aged 21–24 and $7.25 per hour in the U.S., as well as no minimum wage laws in Singapore, the expected payoff of £10 is not a small fee for spending less than 40 minutes in an experiment.

The Golden Balls game is a variant of the prisoner's dilemma with a few crucial differences. First, players are allowed to communicate to each other about what choice they plan to make as a pair, but any agreement they make will be non-binding, unverifiable, and cost nothing if people want to renege on their agreement. Second, although the game requires each player to decide between cooperation or defection, the game's setup looks more like a hawk-dove (or 'chicken') game than a typical prisoner's dilemma game. Like the prisoner's dilemma game, the split-split strategy, while welfare-maximizing, is unstable. However, if one player defects (or steals) in the prisoner's dilemma game, the other player is better off defecting than cooperating (or splitting). By contrast, in the Golden Balls game, if one player chooses to steal, neither player has a better strategy because all strategies that involve stealing are Nash equilibria. In other words, the Golden Balls game is an anti-coordination game where it is mutually beneficial for each player to choose a different strategy.

There are several reasons why we chose the Golden Balls game and not the classic form of the prisoner's dilemma game for our experiments. First, while many of the students across

| | | Player B | |
|---|---|---|---|
| | | Split | Steal |
| **Player A** | Split | £10, £10 | £0, £20 |
| | Steal | £20, 0 | £0, £0 |

**Fig 1. Golden Balls game payoff matrix.**

different disciplines, including economics, psychology, politics, and sociology, would have recently learned about the classic form of the prisoner's dilemma and its dominant strategy from their university classes before entering our lab experiments, they were much less likely to have watched or remembered the Golden Balls game, which was only broadcasted in the late afternoon in the U.K. between 2007 and 2009. Given the existing evidence that some economics training at the university level inhibits cooperation in well-known social dilemma games such as the standard prisoner's dilemma and ultimatum games [22], we intended to present our experimental subjects with a new social dilemma game that is likely to be unfamiliar to most people. Second, we would like to contribute to the small but growing literature on cooperation in prisoner's dilemma games in which defection is a weakly dominant strategy. Although we are interested in studying cooperation, our main goal is to test whether allowing misrepresentation of identity encourages more defection, on average. Since previous experiments show that the cooperation rate tends to be higher in a prisoner's dilemma with weakly dominant strategies than in one with strictly dominant strategies [1], a prisoner's dilemma with weakly dominant strategies such as the Golden Balls provides a better scenario to test whether context can increase the probability of individuals choosing the weakly dominant strategy, i.e., defection. Finally, we believe that a prisoner's dilemma with defection being the weakly dominant strategy and a pre-play communication resembles real interactions between anonymous individuals in cyberspace, where gender misrepresentation is the likeliest place to happen the most. This is because most interactions in cyberspace involve communication between two people in scenarios where each party expects the other to cooperate—e.g., two strangers meeting on Tinder—and that defecting is not strictly better than cooperating under these settings. In addition, the decision to have communication before decisions provide players with an opportunity to explicitly pre-agree with each other on how to behave in the Golden Ball game, thus enabling us to test whether people who misrepresent their gender are still likely to defect and, in effect, break the explicitly-stated agreement in the communication task.

Also, we are not the first study to use prisoner's dilemma games in which defection is a weakly dominant strategy in an experimental setting. For the existing field evidence on prisoner's dilemma games with a weakly dominating strategy, see the real Golden Balls game on TV [32, 33] and gameshows that share the same prisoner's dilemma element as the Golden Balls game, including the American's *Friend or Foe* and the Dutch's *Deelt it 't of deelt ie lt niet*? (English translation: *Will He Share or Not*?) [34–36].

In our Golden Balls game setup, we allowed players 2 minutes to communicate verbally and anonymously via an online messenger, like Facebook messenger. Using Wordfish to conduct a simple text analysis, we find that the most frequently used word during the chat across all treatments and within each treatment was "split," which suggests that cooperation is the intended signal that most people sent to each other, regardless of whether it was followed through. By contrast, there were no significant mentions of words that are related to gender or misrepresentation in the communication; for the text analysis results, see S1 Fig.

There were 143 players in the blind treatment who played the game without any information about the other member. There were 224 players in the true gender treatment who were told the truth about the gender of the other member in the pair before they had to make the split or steal decision. Although we consider the blind treatment as our control group, we also consider the true gender treatment as an alternative reference group as it represents a scenario where each player holds some real information about each other's identity.

There were 332 players in the randomly assigned opportunity to misrepresent treatment. Of those, 166 were randomly assigned an opportunity to misrepresent their gender to the other member. They had to decide after finding out the other member's gender. The other half were explicitly told that the other player was allowed to misrepresent their gender but may or

may not take that opportunity. Of those given the opportunity, 45 took it and misrepresented their gender to the other player, while 121 received the opportunity but chose not to misrepresent. Hence, the ratio between the number of people who chose to misrepresent their gender and the total number of people who were given the opportunity is 27.2%.

We also tested whether randomly forcing some individuals to misrepresent their gender affects the group's later cooperation level. For this treatment, we wanted to test the importance of agency, i.e., the freedom to choose whether to misrepresent. Does randomly forcing some people to artificially misrepresent their gender make them more likely to steal in the Golden Ball games? There were three available conditions ($N = 265$) in this randomly assigned gender treatment. For these players, half ($N = 132$) were explicitly told that they would be shown a randomly assigned gender of the other player that may or may not match their real gender, whilst their true gender will be revealed to the other player. Of those who were told that they would be randomly assigned a gender, 47% ($N = 63$) knew that their gender was being randomly misrepresented to the other player, while 53% ($N = 70$) knew that their true gender was being shown to the other player. There were no deceptions made by the experimenters; only half of the players in the randomly assigned opportunity to misrepresent treatment can deceive their co-participant of their true gender.

Participants filled out the 12-item "Dirty Dozen" Dark Triad personality questionnaire [37] to assess personality, which measures narcissism, psychopathy, and Machiavellianism. The measure consists of 4-items for each personality trait on a scale from 1 (strongly disagree) to 9 (strongly agree). We conducted a factor analysis on the twelve variables to derive with three main factor components of the dark personality traits. Items were summed, with higher scores reflecting higher tendencies towards that trait. All personality traits were entered into the regressions as continuous variables. The personality subscales demonstrated good reliability, see S1 Table. Dark triad personality traits did not statistically differ by experimental conditions, see S2 Table.

Risk preferences were evaluated by asking participants how willing they were to take risks on a scale from 1 to 10, where low scores represent risk aversion. Trust was measured by asking participants the following question: "*Generally speaking, would you say that most people can be trusted, or that you can't be too careful in dealing with other people?*". Participants responded with either "*You can't be too careful*" or "*Most people can be trusted*". Participants were also asked to fill out various socio-demographic questions used as control variables, including age, gender, and if they were part of the student lab sample, their academic major.

## Empirical strategy

The general specification has the decision to split or steal in the Golden Balls game as a linear function of experimental conditions and personal characteristics, which can be written as follows

$$S_i = M_i'\beta + X_i'\gamma + \varepsilon_i. \tag{1}$$

Here, Eq (1) assumes that individual $i$ has a latent propensity to choose either split or steal $S_i^*$. However, we do not observe this latent variable, but the actual split or steal decision $S_i$, where $S_i = 0$ if the person chooses to split and $S_i = 1$ if the person chooses to steal. We impose the observation criterion $S_i = 1(S_i^* > 0)$, where $1(.)$ is the indicator function taking the value of 1 if $(S_i^* > 0)$ and 0 otherwise. The vector $M_i'$ represents dummy variables representing different treatments and conditions within-treatment; $X_i'$ indicates personal characteristics; and $\varepsilon_i$ is the random error term. We estimate Eq (1) using a binary probit model. However, given that probit coefficients are hard to interpret, the estimated marginal effects are reported instead in

the results section. Following the recent paper by Kim [38], we decide not to cluster the standard errors at the session level as doing so could produce false positive coefficients. Nonetheless, we have reported clustered standard errors in our pre-print version and the results are qualitatively the same [39].

## Results

Table 1 reports the selected characteristics by location of the experiment. On average, the steal rates were higher amongst participants in the student sample than in the online sample. In all three locations, the average payment was slightly less than £10, which is the fair outcome in the Golden Balls game. Though not reported in the table, 33% received £0 payment, 46% received £10 payment, and 21% received £20 payment from the game. The Prolific sample participants were the most cooperative of all three locations (82%). They also had the lowest misrepresentation rate—i.e., the proportion of participants who were randomly allowed to misrepresent gender and took the opportunity to do so—in the randomly assigned opportunity to misrepresent gender treatment (14%). One explanation for this might be that the Prolific participants are generally older, have higher incomes, and are more likely to be females than the lab participants, primarily university students.

Fig 2 displays the raw data of mean steal decision across treatments. We found that people were generally cooperative—i.e., the average steal rate was less than 50%–in all conditions. The average steal rate ranged from 29.9% in the 'true gender' treatment to 37.3% in the 'randomly assigned opportunity to misrepresent gender' treatment. However, the Kruskal-Wallis equality-of-population rank test found statistically insignificant group differences in stealing behaviour between the randomised opportunity to misrepresent and the blind treatments $\left(\chi_1^2 = 0.230, \ p = 0.632\right)$, and the randomised opportunity to misrepresent and the true gender treatments $\left(\chi_1^2 = 1.013, \ p = 0.314\right)$.

However, this does not necessarily mean that the treatment effects are not robust. One possible explanation for the statistically insignificant treatment effects in the raw data is that the treatment effect of randomly allowing participants to misrepresent gender may depend on several factors, including the participant's and co-participant's genders and the compliance rate that can vary across locations and the nature of the experiments, i.e., lab versus online. To illustrate this point, Figs 3 and 4 present the raw data of mean steal decision by treatments over

**Table 1. Selected characteristics by location of the experiment.**

| Variables | SG-Lab | UK-Lab | UK/US-Online | Overall |
|---|---|---|---|---|
| Steal (= 1) | 0.36 | 0.43 | 0.18 | 0.33 |
|  | (0.03) | (0.03) | (0.02) | (0.02) |
| Payment | 8.68 | 8.17 | 9.82 | 8.82 |
|  | (0.40) | (0.44) | (0.35) | (0.23) |
| Male | 0.51 | 0.51 | 0.35 | 0.47 |
|  | (0.03) | (0.03) | (0.03) | (0.02) |
| Age | 22.18 | 20.96 | 29.89 | 23.96 |
|  | (0.10) | (0.20) | (0.57) | (0.22) |
| Economics as major | 0.08 | 0.12 | N/A | 0.07 |
|  | (0.01) | (0.02) |  | (0.01) |
| Decided to misrepresent gender in the randomly assigned opportunity to misrepresent treatment | 0.33 (0.07) | 0.39 (0.07) | 0.14 (0.04) | 0.27 (0.03) |
| N | 348 | 338 | 278 | 964 |

**Note**: The mean standard errors are in parentheses.

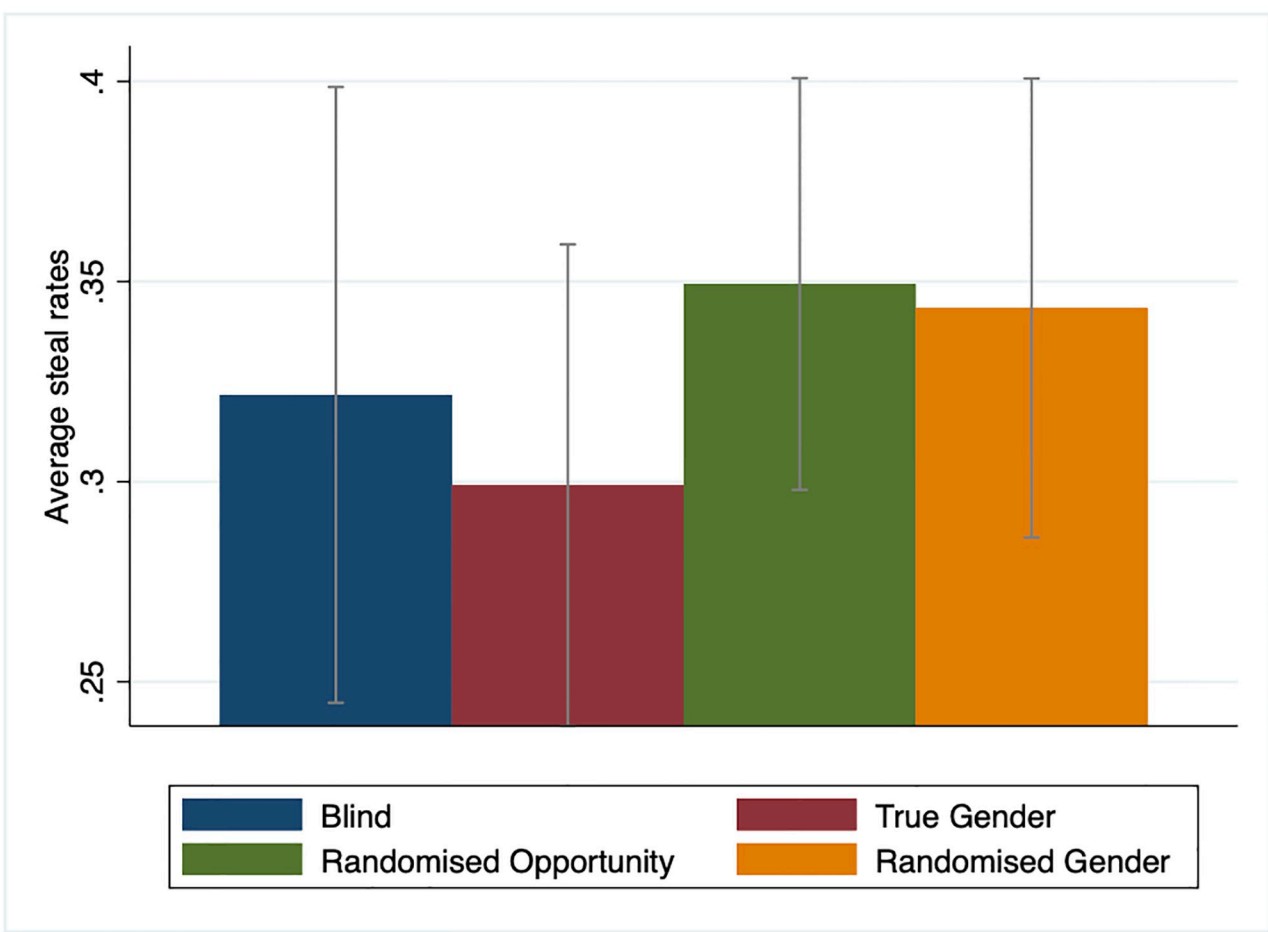

**Fig 2. Average steal rates in the Golden Balls game by treatment.** Blind treatment = information on each participant's gender is not revealed; True gender = information on each participant's gender is revealed to each other in a pair; Randomised opportunity = one participant in each pair is randomly allocated an opportunity to misrepresent their gender, whilst the other participant is told that the other person may or may not be misrepresenting their gender; Randomised gender = one participant in a pair is randomly allocated a gender, which may or may not match their true gender. Error bars represent 95% confidence intervals.

gender-pairing and location of the experiments. Here, we can see from Fig 3 that the mean steal decision for females matched with a male partner is significantly lower in the true gender treatment than other females matched with a male partner in the randomised opportunity to misrepresent treatment. However, the same statement does not apply for other gender pairings between the same two treatments. In addition to this, Fig 4 suggests that the treatment effect of randomly assigned opportunity to misrepresent gender on steal is more positive and statistically robust for participants in the U.K. lab experiment and the Prolific sample than in the Singapore lab experiment sample.

To control for potential confounders that arise from gender pairing and the vastly different samples (and other individual differences), Table 2 enters these treatments into a marginal probit regression with either the blind treatment or the true gender treatment as the reference group. The reason for choosing probit over ordinary least squares is because the outcome variable is a binary variable: cooperate (0) or steal (1). We control in the first column of Table 2 for gender pairing, age, age-squared, a dummy for taking economics as a major at the university, the locations of the experiment, the amount of time taken measured in seconds in the Golden

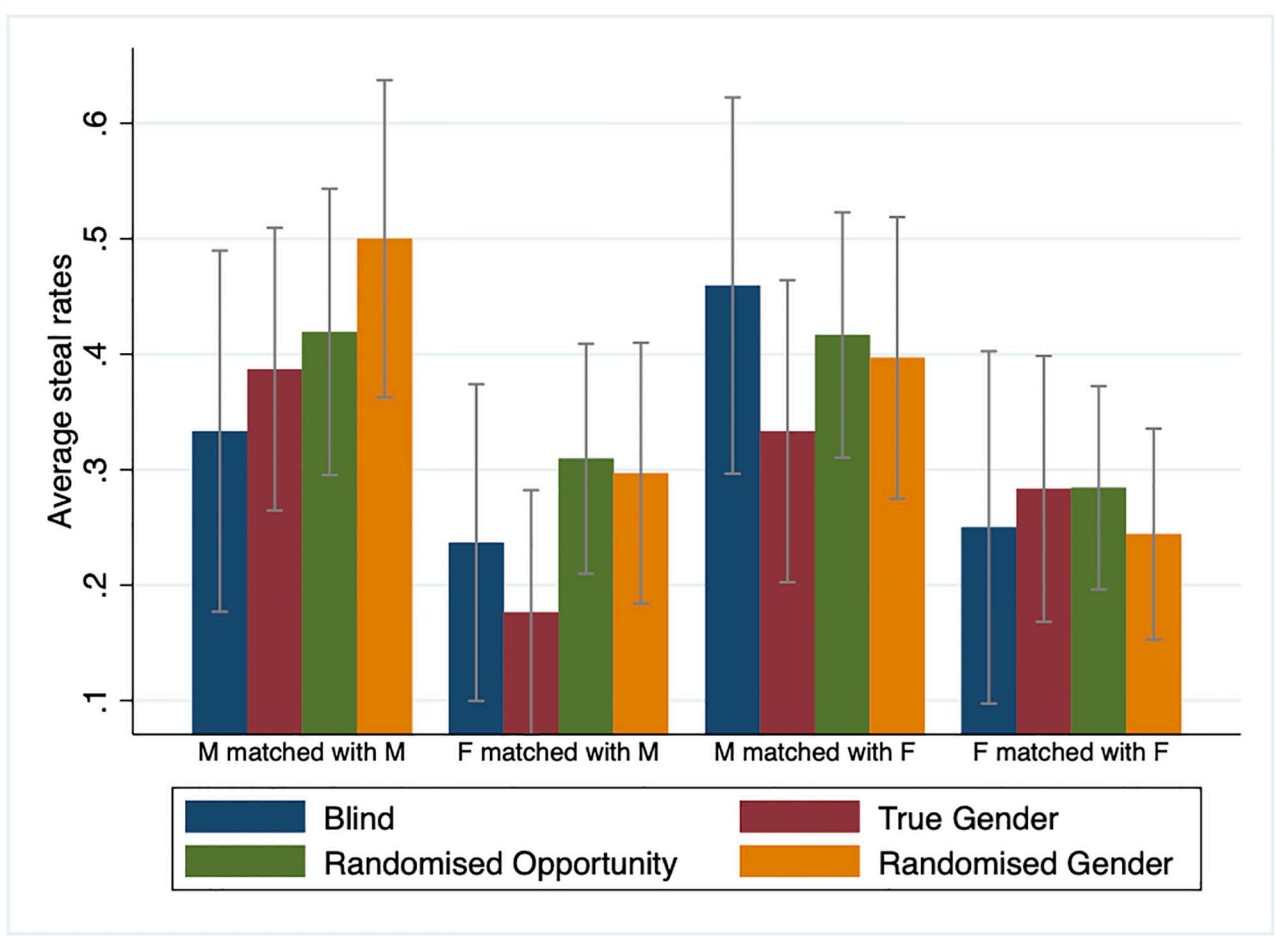

**Fig 3. Average steal rates in the Golden Balls game by treatment and gender pairing.** See Fig 2. M = male participant; F = female participant. Error bars represent 95% confidence intervals.

Balls game, as well as measures of individual differences in attitudes towards risks, the Dark Triad personality traits, and the general trust level.

Looking across the two columns of Table 2, we can see that participants in the randomly assigned opportunity to misrepresent gender treatment are 10.6 percentage points (95% CI: 0.14, 21.0) and 11.6 percentage points (95% CI: 2.72, 20.6) more likely to steal than those in the blind and true gender treatments, respectively. These are sizeable effects, which are approximately half of the effect of holding a belief that no one can be trusted in general.

Regarding the treatment effect of randomly assigned gender on cooperation, we also find some marginal evidence that participants who were randomly assigned a gender that either matched or mismatched their own are 9.89 percentage points (95% CI: -1.02, 20.8) and 10.9 percentage points (95% CI: 1.55, 20.4) more likely to steal than those in the blind and true gender treatments. These estimates suggest that it makes little difference to the average steal rates whether the possibility to misrepresent one's gender in the treatment occurs by choice or by chance; participants in the randomly assigned gender treatment are equally likely to defect as those who had been given agency to misrepresent, on average.

Table 2's other results reveal a similar defection rate between people in the blind treatment and those in the true gender treatment. Women in both mixed-sex and same-sex pairs

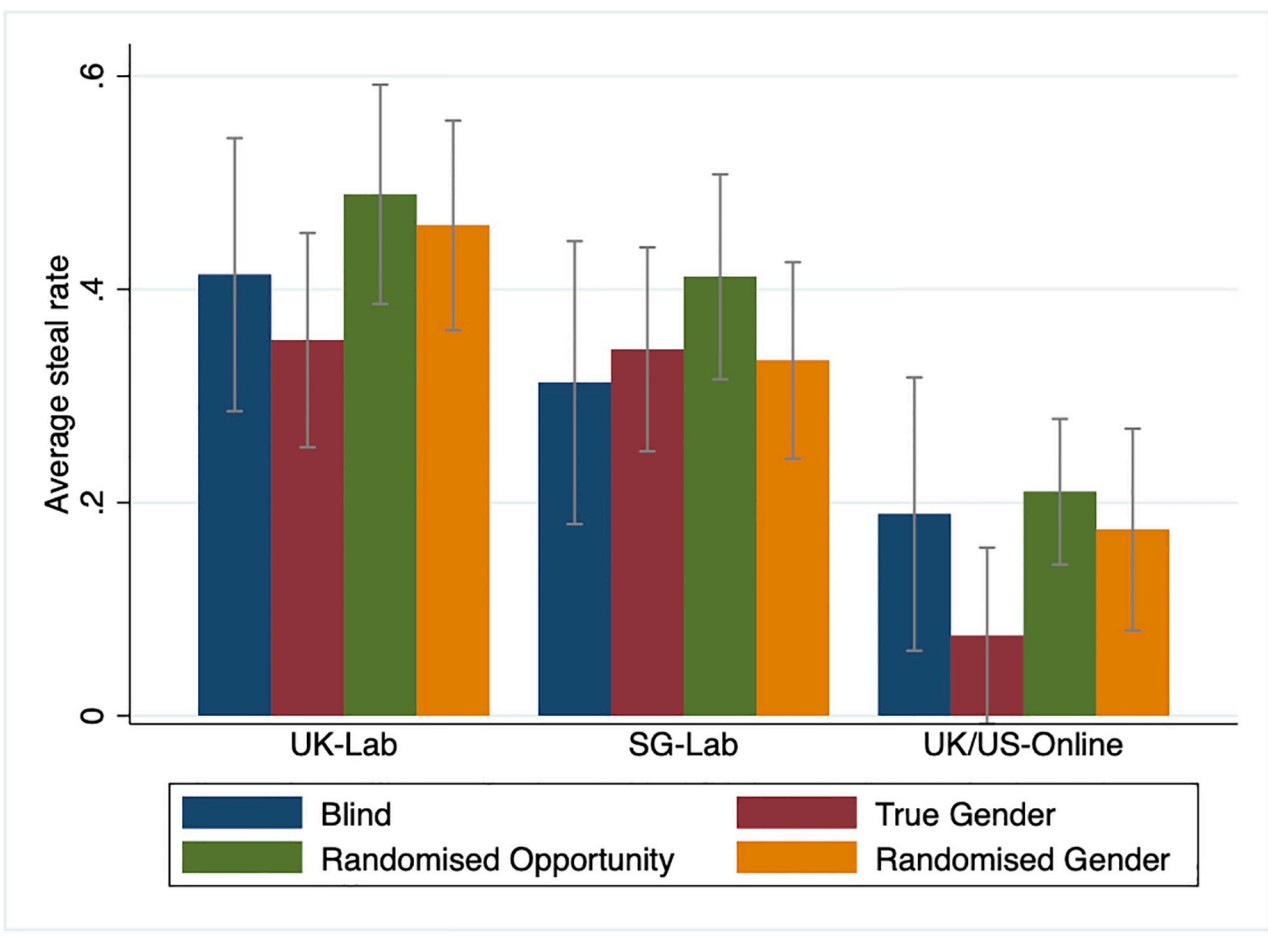

**Fig 4. Average steal rates in the Golden Balls game by treatment and location.** See Fig 2. Error bars represent 95% confidence intervals.

steal substantially less than men in both mixed-sex and same-sex pairs, on average. These results are somewhat consistent with previous studies that find men in same-sex pairs to be the least cooperative of all gender pairings [3, 9]. The steal rates decline with age, although there is a slight uptick among the older participants. There is a negative and statistically significant correlation between the decision to steal and the time it takes to decide the Golden Balls game. We also find that the average steal rates are lowest among online participants even when age and gender are controlled for in the estimation. Concerning the variables on attitudes and personality traits, we find that more narcissistic people are not statistically significantly more likely to steal, whilst those with a high level of the psychopathy and Machiavellianism traits are substantially more likely to steal in the Golden Balls game. Finally, we show that trust is associated negatively and statistically significantly with the tendency to defect, on average.

Other than the effect on own behaviour, another important question is how different treatments affect the behaviour of others. To test this, Table 3 estimate a multinomial probit model on an outcome variable that takes the following values: 0 = both participants split; 1 = participant $i$ splits while the other steals; 2 = participant $i$ steals while the other splits; and 3 = both steal. Note that we cannot directly interpret the multinomial probit coefficients as marginal effects; See S3 Table for the associated marginal effects.

**Table 2. The treatment effects on the decision to steal: Marginal effects probit estimator.**

| VARIABLES | (1) | (2) |
|---|---|---|
| True gender | -0.0102 | |
| | (0.0537) | |
| Blind | | 0.0103 |
| | | 0.0103 |
| Randomised opportunity to misrepresent gender | 0.106** | 0.116** |
| | (0.0533) | (0.0456) |
| Randomised gender | 0.0989* | 0.110** |
| | (0.0557) | (0.0481) |
| **Gender pairing** | | |
| Female matched with male | -0.112*** | -0.112*** |
| | (0.0428) | (0.0428) |
| Male matched with female | 0.0139 | 0.0139 |
| | (0.0469) | (0.0469) |
| Both females | -0.0806* | -0.0806* |
| | (0.0439) | (0.0439) |
| **Personal characteristics** | | |
| Age | -0.0407*** | -0.0407*** |
| | (0.0136) | (0.0136) |
| Age-squared | 0.00054*** | 0.00054*** |
| | (0.000195) | (0.000195) |
| Take Economics as major (if student) | 0.0872 | 0.0872 |
| | (0.0726) | (0.0726) |
| Singaporean sample | -0.0405 | -0.0405 |
| | (0.0375) | (0.0375) |
| Prolific (U.K. and U.S.) sample | 0.697*** | 0.697*** |
| | (0.126) | (0.126) |
| Time taken in the Golden Balls game | -0.0108*** | -0.0108*** |
| | (0.00216) | (0.00216) |
| Risk taking attitudes | 0.00833 | 0.00833 |
| | (0.00754) | (0.00754) |
| Dark triad component: Narcissism | -0.00238 | -0.00238 |
| | (0.0180) | (0.0180) |
| Dark triad component: Psychopathy | 0.0608*** | 0.0608*** |
| | (0.0201) | (0.0201) |
| Dark triad component: Machiavellianism | 0.141*** | 0.141*** |
| | (0.0204) | (0.0204) |
| General trust | -0.261*** | -0.261*** |
| | (0.0298) | (0.0298) |
| Log pseudolikelihood | -476.64 | -476.64 |
| Observations | 963 | 963 |

Note:

*<10%;

**<5%;

***<1%.

Robust standard errors are reported in parentheses. Dependent variable is a binary variable: 0 = split, 1 = steal. The marginal effects are estimated at the means. Note that one person in the online sample got logged out before completing the post-questionnaire.

**Table 3. The treatment effects on paired decision-making: Multinomial probit estimator.**

| VARIABLES | Base outcome: Both split | | |
|---|---|---|---|
| | **Split while the other steal** | **Steal while the other split** | **Both steal** |
| True gender | -0.0321 | 0.126 | -0.360 |
| | (0.224) | (0.233) | (0.288) |
| Randomised opportunity to misrepresent gender | 0.282 | 0.401* | 0.607** |
| | (0.214) | (0.224) | (0.263) |
| Randomised gender | -0.00140 | 0.287 | 0.476* |
| | (0.221) | (0.231) | (0.265) |
| **Gender pairing** | | | |
| Female matched with male | -0.171 | -0.608*** | -0.282 |
| | (0.198) | (0.215) | (0.239) |
| Male matched with female | -0.669*** | -0.144 | -0.395* |
| | (0.205) | (0.205) | (0.235) |
| Both females | -0.561*** | -0.521** | -0.580** |
| | (0.198) | (0.208) | (0.240) |
| **Personal characteristics** | | | |
| Age | 0.0868 | -0.129** | -0.0495 |
| | (0.0784) | (0.0586) | (0.0827) |
| Age-squared | -0.00119 | 0.00178** | 0.000362 |
| | (0.00124) | (0.000830) | (0.00132) |
| Take Economics as major (if student) | -0.633** | 0.0736 | -0.0204 |
| | (0.293) | (0.287) | (0.303) |
| Singaporean sample | -0.514*** | -0.262 | -0.535*** |
| | (0.165) | (0.171) | (0.189) |
| Prolific (U.K. and U.S.) sample | -0.589 | 2.382*** | 1.843** |
| | (0.789) | (0.726) | (0.791) |
| Time taken in the Golden Balls game | -0.00816 | -0.0383*** | -0.0449*** |
| | (0.00964) | (0.00867) | (0.00886) |
| Risk taking attitudes | -0.0326 | 0.0372 | -0.0152 |
| | (0.0295) | (0.0333) | (0.0384) |
| Dark triad component: Narcissism | -0.166** | -0.0736 | -0.0593 |
| | (0.0735) | (0.0803) | (0.0935) |
| Dark triad component: Psychopathy | 0.197** | 0.211** | 0.412*** |
| | (0.0925) | (0.0899) | (0.102) |
| Dark triad component: Machiavellianism | -0.0675 | 0.499*** | 0.394*** |
| | (0.0915) | (0.0883) | (0.104) |
| General trust | -0.531*** | -1.186*** | -1.179*** |
| | (0.140) | (0.157) | (0.182) |
| Log pseudolikelihood | | -1025.207 | |
| Observations | | 963 | |

Note:

*<10%;

**<5%;

***<1%.

Robust standard errors are reported in parentheses. Dependent variable is a categorical variable: 0 = both split; 1 = split, while the other steal; 2 = steal, while the other split; and 3 = both steal. The reported multinomial probit coefficients are not marginal effects. Note that one person in the online sample got logged out before completing the post-questionnaire.

**Table 4. Unpacking the treatment effects on the decision to steal: Marginal effects probit estimator.**

| VARIABLES | All | Female | Male |
|---|---|---|---|
| **True gender** | -0.00937 | -0.0308 | 0.0345 |
| | (0.0535) | (0.0633) | (0.0863) |
| **Randomly assigned opportunity to misrepresent gender** | | | |
| i) Did not receive opportunity to misrepresent | 0.0963 | 0.0954 | 0.0940 |
| | (0.0619) | (0.0778) | (0.0958) |
| ii) Randomly assigned opportunity, did not misrepresent | 0.0395 | 0.0791 | -0.0186 |
| | (0.0668) | (0.0850) | (0.102) |
| iii) Misrepresented gender | 0.319*** | 0.323** | 0.341*** |
| | (0.103) | (0.162) | (0.131) |
| **Randomly assigned gender** | | | |
| i) Were not randomly assigned gender | 0.0909 | 0.0453 | 0.160 |
| | (0.0659) | (0.0823) | (0.0984) |
| ii) Randomly assigned gender/matched | 0.153* | 0.192* | 0.0872 |
| | (0.0813) | (0.103) | (0.127) |
| iii) Randomly assigned gender/mismatched | 0.0603 | 0.113 | -0.00141 |
| | (0.0857) | (0.116) | (0.121) |
| **Wald tests of equality:** $\beta$(Misrepresented gender) *versus* | | | |
| $\beta$(True gender) | $p < .001$ | $p < .010$ | $p < .027$ |
| $\beta$(Randomly assigned opportunity, did not misrepresent) | $p < .009$ | $p < .110$ | $p < .018$ |
| $\beta$(Did not receive opportunity to misrepresent) | $p < .031$ | $p < .134$ | $p < .086$ |
| Other control variables as in Table 2 | Yes | Yes | Yes |
| Log pseudolikelihood | -471.90 | -233.44 | -233.12 |
| Observations | 963 | 517 | 446 |

Note:

*<10%;

**<5%;

***<1%.

Dependent variable is a binary variable: 0 = split, 1 = steal. The marginal effects are estimated at the means using the marginal effects probit estimator. Note that one person in the online sample got logged out before completing the post-questionnaire.

Compared to the blind treatment, the likeliest outcome for participants in the randomised opportunity to misrepresent gender treatment is that both partners in a pair have chosen to independently steal, which would have resulted in both losing the money. There is also some evidence of a successful defection, i.e., steal with the other split, among participants in the randomised opportunity to misrepresent gender treatment, but the estimated coefficient is only marginally statistically significant at the 10% level. In addition, there is also some marginal evidence of participants in the randomised gender treatment choosing to independently steal in any given pair. What these estimates seem to suggest is that randomly allowing people to misrepresent their gender identity is likely to reduce social welfare in a social dilemma situation where cooperation produces the most socially desirable outcome.

We analyse our data further by unpacking the treatment effects in compliers and non-compliers in Table 4. Looking at the entire sample, we find in Column 1 of Table 3 that people who choose to misrepresent their gender are 31.9 percentage points (95% CI: 11.7–52.2) more likely to steal than those in the blind treatment. This is a sizeable coefficient; it is almost twice as large as the effect of not trusting others on the propensity to steal in the Golden Balls game. A

Wald test of equality also suggests that the misrepresented gender coefficient is statistically significantly different from the true gender treatment.

Interestingly, we find that people who were randomly assigned a gender that matched their own are approximately 15 percentage points more likely to steal compared to those in the blind and the true gender treatments, although the estimated coefficient is only marginally statistically significant at the 10% level. One possible explanation for this might be that the random assignment of gender treatment generates a context in which participants are generally less trusting of each other. However, it might also be the case that, for some participants who felt that they had been forced to misrepresent their gender against their will, the righteous thing to do is to overcompensate and defect less than they would have otherwise.

There is weak statistical evidence that the possibility of being paired with someone who might be misrepresenting their gender, whether by choice or by chance, heightens the probability that the individual will defect. Compared to participants in the blind treatment, those who did not receive the opportunity to misrepresent in the randomly assigned opportunity to misrepresent gender treatment are 9.6 percentage points (95% CI: -2.5, 21.7) more likely to steal than individuals in the blind treatment. On the other hand, the effect of playing against someone who might be misrepresenting their gender on defection is much more positive and statistically significant when compared to participants in the true gender group; those who did not receive an opportunity to misrepresent gender are 10.6 percentage points (95% CI: -0.1, 21.4) more likely to defect than those in the true gender treatment. The same applies to those who were not randomly assigned gender in exogenous gender treatment. These findings suggest that the uncertainty of being strategically lied to about the other participant's gender has a potential to lower the average cooperation among those who were not handed an opportunity to misrepresent as well.

The next two columns of Table 4 split the sample by gender. Here, we find that, conditioning on misrepresentation, women and men are 32.3 (95% CI: 5.8, 63.9) and 34 (95% CI: 8.4, 59.8) percentage points more likely to defect than those in the blind group, on average. Hence, Table 4's results provide supporting evidence that most people who chose to misrepresent their gender did so because of the belief that the act of misrepresentation will maximise their chance of a successful defection.

However, Table 5, which reports the multinomial probit regression results on paired decision-making, shows that one of the likeliest outcomes from one partner deciding to misrepresent their gender is that both partners in a pair independently chose to steal. The other similarly probable outcome for the misrepresented gender group is the 'steal while the other split' decision. What Table 5's results seem to imply is that the decision to misrepresent gender tends to precede both partners acting uncooperatively, which caused them both to lose all the money from the pot. Again, since the multinomial probit coefficients are not directly interpretable, we report Table 5's marginal effects in S4 Table.

Table 6 unpacks the treatment effects further by re-estimating Table 4's specification by gender pairing. Note that out of 45 participants who chose to misrepresent their gender, 11 were males in same-sex pairs, 10 were females in mixed-sex pairs, 16 were males in mixed-sex pairs, and 8 were females in same-sex pairs. While there is some evidence that men (N = 27) chose to misrepresent more than women (N = 18), we are not able to reject the null hypothesis that the propensity to misrepresent is statistically significantly between men and women across different gender pairings. In other words, we do not have evidence to support the hypothesis that women in mixed-gender pairing are significantly less likely to want to misrepresent their gender as males because of the perception that men in same-gender pairs are the ones with the most negative expectations about each other's behaviour in a social dilemma interaction [9]. We also estimate a misrepresentation regression to test whether there are any important

**Table 5. Unpacking the treatment effects on the paired decision-making: Multinomial probit estimator.**

| | Base outcome: Both split | | |
| --- | --- | --- | --- |
| | **Split while the other steal** | **Steal while the other split** | **Both steal** |
| **True gender** | -0.0323 | 0.129 | -0.356 |
| | (0.224) | (0.232) | (0.288) |
| **Randomly assigned opportunity to misrepresent gender** | | | |
| i) Did not receive opportunity to misrepresent | 0.340 | 0.353 | 0.630** |
| | (0.243) | (0.252) | (0.297) |
| ii) Randomly assigned opportunity, did not misrepresent | 0.227 | 0.150 | 0.364 |
| | (0.260) | (0.284) | (0.318) |
| iii) Misrepresented gender | 0.189 | 1.078*** | 1.084** |
| | (0.431) | (0.398) | (0.452) |
| **Randomly assigned gender** | | | |
| i) Were not randomly assigned gender | 0.0906 | 0.247 | 0.554* |
| | (0.258) | (0.269) | (0.304) |
| ii) Randomly assigned gender/matched | -0.102 | 0.467 | 0.474 |
| | (0.319) | (0.313) | (0.360) |
| iii) Randomly assigned gender/mismatched | -0.0982 | 0.122 | 0.315 |
| | (0.304) | (0.357) | (0.387) |
| Other control variables as in Columns 3 and 4 in Table 1 | | Yes | |
| Log pseudolikelihood | | -1019.55 | |
| Observations | | 963 | |

Note:

*<10%;

**<5%;

***<1%.

Robust standard errors are reported in parentheses. Dependent variable is a categorical variable: 0 = both split; 1 = split, while the other steal; 2 = steal, while the other split; and 3 = both steal. The reported multinomial probit coefficients are not marginal effects. Note that one person in the online sample got logged out before completing the post-questionnaire.

gender differences in the decision to misrepresent. However, based on the estimates obtained from the decision to misrepresent regression in S5 Table, there is little evidence that men choose to misrepresent more than women in either mixed- or same-sex pair.

By splitting the subsample even further by gender pairing, we find that women in the same-sex pairs who misrepresented themselves as men are 70.4 percentage points (95% CI: 46.1, 94.7) more likely to steal than those in the blind group, on average. Despite the small number of women misrepresented themselves when playing against other women, this is a staggeringly high proportion. We also find some evidence that men who misrepresented themselves as women in the mixed-sex pairs are 29.3 percentage points (95% CI: -5.3, 64.1) more likely to steal than other male counterparts in the blind group. However, the estimate is only marginally significant at the 10% level. These results suggest that men believe that women will cooperate with them more if they can strategically misrepresent themselves as a woman. By contrast, women believe that other women will be nicer to them if they can strategically misrepresent themselves as a man. However, given the small sample size in these subsample regressions, care must be taken when interpreting these marginal effects.

Overall, our results suggest that the possibility of randomly allowing some people to misrepresent their gender to the other person has the potential to lower the average cooperation rates of the entire group substantially.

**Table 6. Misrepresentation and the propensity to steal by gender pairing.**

| VARIABLES | Women in same-sex pairs | Men in same-sex pairs | Women in mixed-sex pairs | Men in mixed-sex pairs |
|---|---|---|---|---|
| **True gender** | 0.0936 | 0.0313 | -0.128* | -0.00919 |
|  | (0.112) | (0.121) | (0.0672) | (0.124) |
| **Randomly assigned opportunity to misrepresent gender** |  |  |  |  |
| i) Did not receive opportunity to misrepresent | 0.159 | -0.000819 | 0.0847 | 0.130 |
|  | (0.126) | (0.140) | (0.103) | (0.131) |
| ii) Randomly assigned opportunity, did not misrepresent | 0.0696 | 0.129 | 0.175 | -0.167 |
|  | (0.124) | (0.167) | (0.132) | (0.111) |
| iii) Misrepresented gender | 0.704*** | 0.266 | -0.131 | 0.293* |
|  | (0.124) | (0.209) | (0.103) | (0.177) |
| **Randomly assigned gender** |  |  |  |  |
| i) Were not randomly assigned gender | 0.0803 | 0.234 | 0.0717 | 0.0486 |
|  | (0.127) | (0.146) | (0.114) | (0.134) |
| ii) Randomly assigned gender/matched | 0.213 | 0.117 | 0.174 | 0.00264 |
|  | (0.145) | (0.197) | (0.152) | (0.157) |
| iii) Randomly assigned gender/mismatched | 0.296 | 0.0163 | -0.0322 | -0.0982 |
|  | (0.208) | (0.168) | (0.0995) | (0.167) |
| **Wald tests of equality:** $\beta$(Misrepresented gender) *versus* |  |  |  |  |
| $\beta$(True gender) | $p < .004$ | $p < .246$ | $p < .020$ | $p < .105$ |
| $\beta$(Randomly assigned opportunity, did not misrepresent) | $p < .005$ | $p < .551$ | $p < .139$ | $p < .015$ |
| $\beta$(Did not receive opportunity to misrepresent) | $p < .013$ | $p < .218$ | $p < .231$ | $p < .397$ |
| Other control variables as in Columns 3 and 4 in Table 1 | Yes | Yes | Yes | Yes |
| Log pseudolikelihood | -115.15 | -112.49 | -99.40 | -112.90 |
| Observations | 280 | 211 | *237* | 235 |

Note:

*<10%;

***<1%.

Dependent variable is a binary variable: 0 = split, 1 = steal. The marginal effects are estimated at the means using the marginal effects probit estimator. Note that one person in the online sample got logged out before completing the post-questionnaire.

## Discussions

We begin our experimental study by asking a question that is more relevant today than ever before: What happens to human cooperation if people are allowed to misrepresent their gender to each other? By randomly allowing half of the participants in treatment to misrepresent their gender in a variant of the prisoner's dilemma game with communication, we increase the average defection rate for the entire group by approximately 12 percentage points from the baseline. This result is driven primarily by the evidence that people who chose to misrepresent their gender strategically were roughly 32 percentage points more likely to defect than those in the baseline group.

This paper's results have both positive and normative implications. First, the possibility of misrepresenting one's gender opens another research avenue for researchers to study how strategies in collective action problems might have to evolve to accommodate the new feature in modern communication technology that allows people to misrepresent their identity on a social network website. Second, our results might explain why dishonest and uncooperative behaviours are widespread on social media and online. According to the Federal Trade Commission (FTC) in America, there were over 25,000 victims who filed a report about romance

scams in 2019, with a total loss of $201 million going to the scammers. Although our study focuses on the effect of gender misrepresentation on cooperation, the overarching implication of our findings is that people can potentially strategically misrepresent any information about themselves that they believe will help nudge the other party to cooperate later in a social dilemma interaction. People may use misrepresentation to gain an advantage from stereotypes such as ethnicity and beauty. For example, people may have the incentive to misrepresent their true physical attractiveness to win other people's trust—as people often do on social media, a behaviour commonly known as 'catfishing'–because of the existing stereotypical belief that more beautiful people are more trustworthy, more cooperative, and better negotiators (e.g., Mobius & Rosenblat, 2006; Andreoni & Petrie, 2008; Rosenblat, 2008), thereby maximising their chance of a successful defection later. Future research should investigate the implications of other types of misrepresentation across different social dilemma games.

Furthermore, an important policy implication of our results comes from the finding that people in the true gender treatment are statistically the most cooperative of all groups. This suggests that we may be able to curb dishonest and uncooperative behaviours online by allowing people to verify and authenticate their social media profiles. However, no such policy is currently in place in any of the major social media platforms. Future research may have to return to evaluate whether a large-scale enrolment of opportunities for people to authenticate their online presence can effectively reduce the incidence of uncooperative behaviours among internet users.

Like all papers in social sciences, our study is not without limitations. One concern is the generalisability of our findings. We take this opportunity to express what we believe to be the constraints on the generality of our results [40]. The current study shows that randomly allowing people to misrepresent their gender substantially lowers the average cooperation levels in a variant of the prisoner's dilemma game with communication for the entire group. While we do not have evidence that the findings will be reproducible for other types of strategic games, such as trust games, public goods games, and dictator games, we believe that our results are generalisable in settings where there is a randomised opportunity for people to misrepresent.

Given that most, if not all, of our student participants, use social media daily, we also believe the results will be reproducible with a sample of randomly selected social media users across different countries. However, whether our results can be generalised to scenarios where the stakes are large and in repeated interactions remains to be seen. Another shortcoming is that we did not elicit subjects' beliefs about the interaction between gender and cooperation, which limits our understanding of the mechanisms that might be guiding our results. Also, future research may have to come back to investigate whether the results will be replicable had the classical Prisoner's Dilemma been used instead of the Golden Balls game. Finally, we have no reason to believe that the results depend on other characteristics of the subjects, materials, or context that are not already accounted for in the current study.

## Supporting information

**S1 Table. Descriptive statistics for study variables.** Sample α displays the samples Cronbach's alpha for multi-item measures. Economics major excludes participants from the Prolific sample as they were not recruited from a student sample, 0 indicates student participants are studying a subject which is not economics, and 1 indicates student participants are studying economics. Steal is a dummy variable where 0 indicates selecting split and 1 indicates selecting steal. Trust is a dummy variable where 0 indicates participants selected "You can't be too careful" and 1 indicates participants selected "Most people can be trusted".
(DOCX)

**S2 Table. Dark triad personality by experimental condition.** One-way ANOVAs indicate there are no statistical differences in dark triad traits between groups.
(DOCX)

**S3 Table. Marginal effects obtained from Table 3's multinomial probit model.** \*\*<5%. Blind treatment is the reference group.
(DOCX)

**S4 Table. Marginal effects obtained from Table 4's multinomial probit model.** \*\*<5%; \*<10%. Blind treatment is the reference group.
(DOCX)

**S5 Table. Marginal effects from probit regression on the decision to misrepresent.** \*<10%; \*\*\*<1%. Heteroskedasticity-adjusted standard errors at the sessional level are reported in parentheses. The sample consists of those who were randomly allowed the opportunity to misrepresent. Dependent variable is a binary variable: 0 = received an opportunity to misrepresent but did not take it, 1 = misrepresented. The marginal effects are estimated at the means.
(DOCX)

**S1 Fig. Text analysis of the 2-minute communication across all treatments.** Text analysis on the 2-minute communication between players across all treatments shows that "split" is the most frequently used words in the conversation. The larger the words are in this word cloud, the more frequent they appeared in the conversation.
(DOCX)

## Acknowledgments

We are grateful to Ta Vejpattarasiri and Erwin Wong for their help with the data collection. We are also thankful to Andrew Oswald, Carol Graham, and Daniel Sgroi for their excellent comments on the draft. The experiment was approved by the HSSREC ethics board at the University of Warwick (Ref: H.S.S. 75/18-19).

## Author Contributions

**Conceptualization:** Michalis Drouvelis, Jennifer Gerson, Nattavudh Powdthavee, Yohanes E. Riyanto.

**Data curation:** Michalis Drouvelis, Jennifer Gerson, Yohanes E. Riyanto.

**Formal analysis:** Jennifer Gerson, Nattavudh Powdthavee.

**Investigation:** Nattavudh Powdthavee, Yohanes E. Riyanto.

**Methodology:** Michalis Drouvelis, Jennifer Gerson, Nattavudh Powdthavee, Yohanes E. Riyanto.

**Software:** Michalis Drouvelis.

**Writing – original draft:** Jennifer Gerson, Nattavudh Powdthavee.

**Writing – review & editing:** Michalis Drouvelis, Jennifer Gerson, Nattavudh Powdthavee, Yohanes E. Riyanto.

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
