## [Decision Letter · Decision Letter 0]

14 Nov 2022

PONE-D-22-25168Large Losses from Little Lies: Strategic Gender Misrepresentation and CooperationPLOS ONE

Dear Dr. Powdthavee,

Thank you for submitting your manuscript to PLOS ONE. After careful consideration, we feel that it has merit but does not fully meet PLOS ONE’s publication criteria as it currently stands. Therefore, we invite you to submit a revised version of the manuscript that addresses the points raised during the review process.

We look forward to receiving your revised manuscript.

Kind regards,

Luo-Luo Jiang, Ph.D.

Academic Editor

PLOS ONE

Journal Requirements:

2. Please clarify whether the preprint (posted here as a working paper -https://papers.ssrn.com/sol3/papers.cfm?abstract_id=3699244) has undergone peer-review.

   "We are grateful to Ta Vejpattarasiri and Erwin Wong for their help with the data collection. We are also thankful to Andrew Oswald, Carol Graham, and Daniel Sgroi for their excellent comments on the draft. The project was funded by M.D., N.P., and Y.E.R.’s annual personal research budget from their respective universities. The experiment was approved by the HSSREC ethics board at the University of Warwick (Ref: HSS 75/18-19)."

4. We noted in your submission details that a portion of your manuscript may have been presented or published elsewhere. Please clarify whether this was peer-reviewed and formally published. If this work was previously peer-reviewed and published, in the cover letter please provide the reason that this work does not constitute dual publication and should be included in the current manuscript.

Reviewers' comments:

Reviewer's Responses to Questions

**Comments to the Author**

1. Is the manuscript technically sound, and do the data support the conclusions?

Reviewer #1: No

Reviewer #2: Partly

Reviewer #3: Yes

2. Has the statistical analysis been performed appropriately and rigorously? 

Reviewer #1: No

Reviewer #2: Yes

Reviewer #3: N/A

3. Have the authors made all data underlying the findings in their manuscript fully available?

Reviewer #1: Yes

Reviewer #2: Yes

Reviewer #3: Yes

4. Is the manuscript presented in an intelligible fashion and written in standard English?

Reviewer #1: Yes

Reviewer #2: Yes

Reviewer #3: Yes

5. Review Comments to the Author

Reviewer #1: Summary of the paper

This paper investigates the role of mispresenting gender in determining cooperation in social dilemma. The authors recruit subjects from three different samples and test their idea with a golden ball game experiment. Using four different treatment, the authors found that randomly allowing participants to misrepresent their gender to the other player reduces the aggregate cooperation level for the entire treatment group, and individuals who chose to misrepresent their gender are more likely to defect than those in the control group. The research question is clear and interesting, and the design and implementation of the experiment is reasonable. However, the robustness of the finding is not convincing.

Major comments:

1. The main finding of the paper is that “randomly allowing participants to misrepresent their gender to the other player reduces the aggregate cooperation level for the entire treatment group by approximately 12 percentage points”. The crucial argument is not supported by the data. First, “Randomly assigned opportunity to misrepresent gender” should be one treatment, whether one received the opportunity could not be seen as a treatment design. According to the data, the overall effect of this treatment is not different from the control (31.9% vs. 34.9% steal). Second, even we allow this split, the subsample of “Randomly assigned opportunity to misrepresent gender” is only significant at a 10% level in only the regression that including many other factors (model (3) of table 2). This effect is not significant in model (1). The “approximately 12 percentage points” is based on the coefficient of model (3) while not real overall effect. Third, the authors did not report average date on treatment level but we check the data and did not find any significant difference between treatments. Thus, the main findings are not convincing.

2. The second important finding is “individuals who chose to misrepresent their gender are 32 percentage points more likely to defect than those in the control group”. The problems here are: first, the research question is the impact of mispresenting gender on cooperation. It is not surprise to find those who prefer to lie for their gender also tend to steal in the game. They are just those bad apples. More interesting question is how this affect the behavior of others. Unfortunately, I did not find any analyses on the interactive behaviors between pairs. Second, Even you would like to compare the decision of those who choose to mispresent, you should compare them to those who choose not the mispresent while not to the control treatment.

3. The findings are not in line with the hypotheses. The authors propose that “there may be less incentive for women in mixed-sex pairs to strategically misrepresent their gender to their partner, considering that men in same sex pairs are ones with the most negative expectation about each other’s behaviour in a social dilemma interaction”. However, 10 were females in mixed-sex pairs chose to misrepresent their gender, which is not different with other conditions.

Minor comments:

1. The study use the Golden Balls game but not the prisoner’s dilemma to study cooperation and justify this with “many of the students would have recently learned about the classic form of the prisoner’s dilemma”. It should not be a reason for a study. Why do not you choose samples who did not learn prisoner’s dilemma. Besides, the data suggest that only 7% subjects were from economics.

2. The experiment allowed communication before decisions. This design seems meaningless to the research question. The authors did not justify why you use this design and did nor provide further analyses for this. In addition, the player can learn whether the other side break the agreement from their payoff. So the agreement is not “unverifiable”.

3. The authors did not clearly present their treatment settings. For example, in second paragraph of P.13, the explanation of the three available conditions in randomly assigned gender treatment is confusing.

4. Several writings of the paper are not precise: such as the expression of “was slightly less than 10”, “the average steal rate was less than 50%”. Why do not directly report the exact figure.

5. The paper also lacks a sufficient review of the related literature about lying/cheating, social identity, and social preference.

6. To support the motivation and hypotheses of the study, it is expected the authors could do further measures such as the above mentioned interaction between pairs and subjects’ beliefs.

Reviewer #2: In this study, the authors studied a very interesting question: what would the ability and choice of misrepresenting one's gender affect their propensity to cooperate in social dilemma setting. This is a well-motivated problem especially in the internet era where most online inter-personal interactions are made behind anonymity and self-selected gender identity. To avoid prior knowledge to the prisoner's dilemma problem, they choose the Goldern Balls game, a variant of the prisoner's dilemma and collected experimental results from two countries. Results suggest that the the choice to misrepresent genders in Goldern Balls can lead to significantly more defective behaviors than the control group.

Overall, the experimental setting is relatively comprehensive, and so are the results. The hypotheses are logical after the well-written introduction of related work. The description of the experimental settings and methodologies, however, can benefit from writing improvements.

P9. The description of the four treatments (and later separated into six treatments) are confusing. How are "4b. Was not randomly assigned gender" and "3a. Did not receive a random opportunity to misrepresent gender" different from "1. Blind"? Is it the case that 3a or 4b participants know that their opponents can be gender-misrepresented or randomly assigned gender, but they cannot?

And in Figure 2, abbreviations are used to indicated results, while here in method description, these abbreviations are never introduced or used. Please standardize them.

P12-13. The different conditions are mixed together with the grouping based on these participants responses. It might be much easier for the readers to understand to use a table where the columns and rows factorize these different conditions and responses, and put their corresponding numbers (e.g. number of people who choose to misrepresent, number of people who defects etc.) in it.

P13. "The other half were told that the other player was allowed to misrepresent their gender but may or may not take that opportunity" -- for the other half, are they simply not told this, or are they explicitly told that the other player are forbidden from misrepresenting their gender. This can make a difference, because they might assume the other player are also allowed to misrepresent even if they are not told so.

Following the previous point, if they are simply not told anything. Perhaps an experimental condition can better understand this effect, where that they are explicitly told the other opponent's gender is true.

P13. "The other half were told that the other player was allowed to misrepresent their gender but may or may not take that opportunity. Of those who were given the opportunity, 45 took it and misrepresented their gender to the other player, while 121 received the opportunity but chose not to misrepresent. Hence, the compliance rate is around 27%." -- how do the compliance rates differ for those who are told that their opponents can misrepresent vs. who are told that their opponents cannot.

P13. Unlike in the misrepresentation choice setting above, in the random assignment setting, "In both conditions, the opposing player was told that their opponent’s gender information may not be accurate." Why the difference? Would it make more sense to also consider a setting where their are told that their opponent are not randomly assigned?

The mathematically notations seem to be messed up. There are many empty blocks of rectangles in place of where some symbols should be at (e.g. P12-14). Please fix them.

P14. How are the linear function fitted? Please describe.

Last but not least, there is an effect that the authors didn't study, that may offer an alternative explanation to the results. In the background section, the authors hypothesize that the participants, if given the choice to misrepresent their genders, might take advantage of it by choosing the gender that minimize their opponents' negative expection about themselves. However, this is under the assumption that all the participants are explicitly aware of the gender effect on the social dilemma interactions (e.g. women would have a strong preference to signal to other women that they are kind; and men would cooperate more if their opponents are women). In reality, however, these knowledge are not necessarily conscious to the participants. Therefore, the set of experiments could be additionally separated into two conditions, one where the participants are given a fact sheet of these gender-related cooperation preferences, and one where they aren't. It would be quite interesting to understand this effect.

In summary, the referee would like to see an improvement of the method description and additional experimental results that considered the effect of the awareness of gender preferences, and the opponent's experimental settings. Other than these, the research can illuminate important insights on critical real-world interactions in our digital life.

Reviewer #3: Summary

The paper entitled Large Losses from Little Lies: Strategic Gender Misrepresentation and Cooperation investigates an interesting question on cooperation with the possibility of misrepresenting one’s gender. The lab and the field experiment study provide large-sample and multinational evidence that confirm the harmful impact of the available deception on gender. This paper extends mixed previews findings on gender differences in cooperation and consider gender stereotype. Overall, this paper is interesting and helpful in understanding the relationship between small lies on gender and cooperation. However, I have some concerns which may help during the revision.

Major Comments

1. What are the purpose and settings of the fourth treatment- Randomly assigned gender? Does this treatment involve deception on subjects? Even if not, I think this design is counterfactual and unreasonable. It is almost impossible to force people to misrepresent their gender in reality. So, not surprisingly, the result of the treatment is disappointing. By contrast, the treatments in regard of hidden gender or gender composition of populations make more sense.

2. There might be some exciting findings in the verbal and anonymous communication before the game. Did the participants have the chance to send gender messages in this part? If so, these messages might be part of the effect of gender misrepresentation or bias in the estimation. Secondly, since this communication is unstructured and participants can talk about whatever they want, does the text data reflects some interesting behavior pattern? For instance, did the participants misrepresenting gender grab this chance to convince their partner to believe them? The authors should consider digging deep into the text data.

3. The consistency between lying behaviors and stealing/splitting behavior should be checked. Some participants might lie to take advantage of stereotypes. However, there might be some participants who were afraid of defection. Once they get the chance to misrepresent, they might lie about their gender to protect themselves from others' defection. This type of participant might be more willing to cooperate. The authors should analyze the different behavior patterns further.

4. One of the most interesting results is the different strategies used by male and female. Which strategy is closer to the rational expectation or which is the more successful strategy? Is there any deeper psychological explanation for this result?

5. Last but not least, the regression coefficients of most treatments are not significant, which means the impacts are small statistically. To make conclusions more convincing, maybe more explanations are needed. What’s more, the analysis on mechanism is somewhat too simple and superficial, it should be more in-depth, for example, including people’s beliefs which are crucial to the results.

Minor comments

1. There are several mistakes in the paper. In Introduction section, (p4 line2) "a variant" repeated for 2 times. In Empirical strategy section, p14, paragraph 4, line 3 and 4, letters of the equation failed to be shown, perhaps due to the document export problems.

2. I suggest adding some non-parametric statistical hypothesis tests before skipping into a probit regression, which may help explore differences between two treatments and focus on the significant outcomes. Figure 2 alone is insufficient to clarify the overall treatment effect of four conditions.

3. Wilcoxon Signed Rank in the second paragraph on Page 15 is misused. As mentioned above, the Wilcoxon Rank Sum Test is the test that is used in a between-subject design. It should be checked

4. The reference list are not completed. For example, “Kim (2020)” in the page 14 is not included in the references.

6. PLOS authors have the option to publish the peer review history of their article (what does this mean?). If published, this will include your full peer review and any attached files.

Reviewer #1: **Yes: **Shuguang Jiang

Reviewer #2: **Yes: **Baihan Lin

Reviewer #3: **Yes: **Yefeng Chen

---

## [Author Response · Author response to Decision Letter 0]

2 Jan 2023

Reviewer #1: Summary of the paper

This paper investigates the role of mispresenting gender in determining cooperation in social dilemma. The authors recruit subjects from three different samples and test their idea with a golden ball game experiment. Using four different treatment, the authors found that randomly allowing participants to misrepresent their gender to the other player reduces the aggregate cooperation level for the entire treatment group, and individuals who chose to misrepresent their gender are more likely to defect than those in the control group. The research question is clear and interesting, and the design and implementation of the experiment is reasonable. However, the robustness of the finding is not convincing.

Major comments:

1. The main finding of the paper is that “randomly allowing participants to misrepresent their gender to the other player reduces the aggregate cooperation level for the entire treatment group by approximately 12 percentage points”. The crucial argument is not supported by the data. First, “Randomly assigned opportunity to misrepresent gender” should be one treatment, whether one received the opportunity could not be seen as a treatment design. According to the data, the overall effect of this treatment is not different from the control (31.9% vs. 34.9% steal). Second, even we allow this split, the subsample of “Randomly assigned opportunity to misrepresent gender” is only significant at a 10% level in only the regression that including many other factors (model (3) of table 2). This effect is not significant in model (1). The “approximately 12 percentage points” is based on the coefficient of model (3) while not real overall effect. Third, the authors did not report average date on treatment level but we check the data and did not find any significant difference between treatments. Thus, the main findings are not convincing.

**Firstly, thank you very much for taking the time to read and critically comment on our paper. We recognise how much time and effort went into doing so. We agreed with your comment that the raw data in Fig.1 does not provide sufficient evidence that there is any significant effect from randomly allowing people to misrepresent as stated in the abstract. We also took your advice that we should, at first, be looking at the treatment effect – rather than splitting the sample into i) those who were not given the opportunity, and ii) those who were – as a whole.

Let us begin with an explanation as to why the overall effect of this treatment is not different from the control in Fig.1. As we have now shown in new Figs. 2 and 3, the treatment effects are likely to be confounded by i) gender pairing, and ii) the very different samples that we used. For instance, as shown in Fig.2, giving people information about the other partner’s gender is likely to affect how females and males play in the Golden Ball game compared to the Blind condition where information about each other’s gender is not available. Here, we can see that there is a significant difference in the steal rates among females when they were matched against males between those in the Blind treatment and the True gender treatment. 

Also, as shown in Fig.3, how the experiment was conducted, i.e., lab or online, seems to matter to the defection rate by treatment. While participants in the randomly assigned opportunity to misrepresent gender treatment defected substantially more than those in the Blind treatment in the UK-lab and online, the difference is not as clear cut in SG-lab. While having vastly different samples in an experiment is typically viewed as a plus, it also implies that more care needs to be taken when interpreting raw data variations. 

Nevertheless, we redid our Table 2 and looked at the treatment effects instead of dividing them up. Since we now have a larger N per cell and equipped with the hypothesis that people who were not assigned the opportunity to mispresent are also likely to defect because of the fear that they may be playing against someone who is likely to act deceitfully, Column 1 of Table 2 now reports a positive and statistically robust coefficient of 0.106 (S.E.=0.053), which suggests that participants in the “randomised opportunity to misrepresent gender” are 10.6 percentage points more likely to defect than those in the Blind treatment. In addition to this, we also present in Column 2 of Table 2 the estimated treatment effects with participants in the True Gender treatment as the baseline group. Since participants’ gender in the True Gender treatment is likely to have some implications on people’s cooperative decision, it seems valid and important for us to compare the treatment effects between participants in the randomly assigned opportunity to misrepresent and those in the True Gender treatments as well. Here, we continue to find a strong treatment effect of 0.116 (S.E.=0.045). We also find the treatment effects of exogenously randomised gender to be significant in this specification, which also suggests that giving people uncertainty about the true gender identity, whether it is by choice or not, can also corrode people’s willingness to cooperate as a group. Overall, we can conclude based on Table 2’s estimates that the treatment effects are significantly robust. 

***

2. The second important finding is “individuals who chose to misrepresent their gender are 32 percentage points more likely to defect than those in the control group”. The problems here are: first, the research question is the impact of mispresenting gender on cooperation. It is not surprise to find those who prefer to lie for their gender also tend to steal in the game. They are just those bad apples. 

**We fully appreciate why you might think that “It is not a surprise to find those who prefer to lie for their gender also tend to steal in the game. They are just those bad apples”. However, if that is the case, then we should also see these ‘bad apples’ defect even if they do not have the opportunity to lie about their gender in a similar proportion across all treatments through the process of randomisation, but we did not. The key message of the paper is that it is the difference in context that facilitates these ‘bad applies’ to behave uncooperatively and not their bad personalities, which are equally distributed across treatments. But we acknowledge that this point might not have been made explicitly clear enough in the paper, so we have now added a brief discussion in the revised version to make sure that this message gets across to our readers.

***

More interesting question is how this affect the behavior of others. Unfortunately, I did not find any analyses on the interactive behaviors between pairs. Second, Even you would like to compare the decision of those who choose to mispresent, you should compare them to those who choose not the mispresent while not to the control treatment.

**This is a great comment, which we took seriously and, as a result, estimated two new tables – Tables 3 and 5. These two new tables looked at the treatment effects on paired decision. Here, the possible outcome variables are the following: both participants in a pair split; one split while the other steal; one steal while the other split; and both steal. Using multinomial probit estimator, we show in Table 5 that not only people who misrepresented their gender are more likely to steal, those who played against them are also likely to steal as well. 

3. The findings are not in line with the hypotheses. The authors propose that “there may be less incentive for women in mixed-sex pairs to strategically misrepresent their gender to their partner, considering that men in same sex pairs are ones with the most negative expectation about each other’s behaviour in a social dilemma interaction”. However, 10 were females in mixed-sex pairs chose to misrepresent their gender, which is not different with other conditions.

**This point is well-taken, and we have now explicitly stated on p.22 in the revised manuscript that we did not find such evidence to support the hypothesis that women in mixed-gender pairs are substantially less likely to want to mispresent their gender because of how men perceive each other in a social dilemma game.

Minor comments:

1. The study use the Golden Balls game but not the prisoner’s dilemma to study cooperation and justify this with “many of the students would have recently learned about the classic form of the prisoner’s dilemma”. It should not be a reason for a study. Why do not you choose samples who did not learn prisoner’s dilemma. Besides, the data suggest that only 7% subjects were from economics.

**The reason “many of the students would have recently learned about the classic form of the prisoner’s dilemma” was only one of the reasons why we chose the Golden Ball game. Of course, it was not a priori possible to know how many students are familiar with the Prisoner’s Dilemma (PD) game. Also, the PD game is not only taught in economics, but in politics, psychology, and sociology. Also, given that there have been so many papers that looked at the Golden Ball game as a valid strategic game (see a few papers published in Management Science), it did not occur to us why it would not be possible to continue down the same path as previous studies. In addition to this, the natural set-up of the Golden Ball game allows communication between partners, which we thought it was an attractive feature to have because it allows people to ‘lie’ about their intention following a decision of whether to misrepresent gender.

2. The experiment allowed communication before decisions. This design seems meaningless to the research question. The authors did not justify why you use this design and did nor provide further analyses for this. In addition, the player can learn whether the other side break the agreement from their payoff. So the agreement is not “unverifiable”.

**We have now discussed how having communication before decisions allows players to explicitly pre-agree before deciding what to do in the Golden Balls game. This way, we can see whether people who misrepresent are more likely to break the explicitly-stated agreement. Without communication, players will have to independently assume how the other player would play and no pre-agreements can be made beforehand.

3. The authors did not clearly present their treatment settings. For example, in second paragraph of P.13, the explanation of the three available conditions in randomly assigned gender treatment is confusing.

**We have since revised the sentences in this paragraph to make it less confusing.

4. Several writings of the paper are not precise: such as the expression of “was slightly less than 10”, “the average steal rate was less than 50%”. Why do not directly report the exact figure.

**This has since been rectified.

5. The paper also lacks a sufficient review of the related literature about lying/cheating, social identity, and social preference.

**We are not entirely sure about this suggestion, given that we have already discussed the main paper on social identity and its implications on p.5 (e.g., Croson and Gneezy, 2009) and that the literature on lying/cheating, though related, does not seem to be entirely relevant in this context. We believe that we have been quite comprehensive in discussing the relevant literature on gender and cooperation in strategic games. Nevertheless, we believe that we could reference a few studies that find uncertainty about the opponent’s strategy reduces individual’s willingness to cooperate on p.8. 

6. To support the motivation and hypotheses of the study, it is expected the authors could do further measures such as the above mentioned interaction between pairs and subjects’ beliefs.

**Thank you very much for this suggestion, but this is beyond the scope of this paper, which was also pre-registered. We have, however, put this in the limitation of the study that appears in the conclusion section.

——

Reviewer #2: In this study, the authors studied a very interesting question: what would the ability and choice of misrepresenting one's gender affect their propensity to cooperate in social dilemma setting. This is a well-motivated problem especially in the internet era where most online inter-personal interactions are made behind anonymity and self-selected gender identity. To avoid prior knowledge to the prisoner's dilemma problem, they choose the Goldern Balls game, a variant of the prisoner's dilemma and collected experimental results from two countries. Results suggest that the the choice to misrepresent genders in Goldern Balls can lead to significantly more defective behaviors than the control group.

Overall, the experimental setting is relatively comprehensive, and so are the results. The hypotheses are logical after the well-written introduction of related work. The description of the experimental settings and methodologies, however, can benefit from writing improvements.

P9. The description of the four treatments (and later separated into six treatments) are confusing. How are "4b. Was not randomly assigned gender" and "3a. Did not receive a random opportunity to misrepresent gender" different from "1. Blind"? Is it the case that 3a or 4b participants know that their opponents can be gender-misrepresented or randomly assigned gender, but they cannot?

**Thank you very much. Following Referee 1’s comments, we have now substantially revised the description of the experimental setup in the methodology section to clear these confusions.

And in Figure 2, abbreviations are used to indicated results, while here in method description, these abbreviations are never introduced or used. Please standardize them.

**We have since revised Fig.2 to make the labels easier to understand.

P12-13. The different conditions are mixed together with the grouping based on these participants responses. It might be much easier for the readers to understand to use a table where the columns and rows factorize these different conditions and responses, and put their corresponding numbers (e.g. number of people who choose to misrepresent, number of people who defects etc.) in it.

**We have rewritten how we described the sample (and the sample size in each condition) to make them easier to follow and understand.

P13. "The other half were told that the other player was allowed to misrepresent their gender but may or may not take that opportunity" -- for the other half, are they simply not told this, or are they explicitly told that the other player are forbidden from misrepresenting their gender. This can make a difference, because they might assume the other player are also allowed to misrepresent even if they are not told so.

**Following the previous point, if they are simply not told anything. Perhaps an experimental condition can better understand this effect, where that they are explicitly told the other opponent's gender is true.

They were explicitly told that the other person in a pair was given the opportunity to misrepresent their gender. We have now made this point clearer in the text.

P13. "The other half were told that the other player was allowed to misrepresent their gender but may or may not take that opportunity. Of those who were given the opportunity, 45 took it and misrepresented their gender to the other player, while 121 received the opportunity but chose not to misrepresent. Hence, the compliance rate is around 27%." -- how do the compliance rates differ for those who are told that their opponents can misrepresent vs. who are told that their opponents cannot.

**The compliance rate here applies to the ratio between people who misrepresented gender (45) and those who were given the opportunity (166) = 27%, so we are not sure what you meant by “the compliance rate of those who are told that their opponents cannot” since there was nothing for them to comply to. Perhaps there was a misunderstanding in what the compliance rate means here, so we have reworded it in the text to the ‘ratio’.

P13. Unlike in the misrepresentation choice setting above, in the random assignment setting, "In both conditions, the opposing player was told that their opponent’s gender information may not be accurate." Why the difference? Would it make more sense to also consider a setting where their are told that their opponent are not randomly assigned?

**We did this because we wanted to be fully transparent to both partners what is going on, and do not want to involve any deception in our experiment. 

The mathematically notations seem to be messed up. There are many empty blocks of rectangles in place of where some symbols should be at (e.g. P12-14). Please fix them.

**We checked all the mathematical notations but have found nothing wrong. Perhaps that was due to the PDF conversion provided by PLOS ONE?

P14. How are the linear function fitted? Please describe.

**We have already stated in the paragraph, which might have been missed earlier, that the linear function was fitted using a probit model.

Last but not least, there is an effect that the authors didn't study, that may offer an alternative explanation to the results. In the background section, the authors hypothesize that the participants, if given the choice to misrepresent their genders, might take advantage of it by choosing the gender that minimize their opponents' negative expectation about themselves. However, this is under the assumption that all the participants are explicitly aware of the gender effect on the social dilemma interactions (e.g. women would have a strong preference to signal to other women that they are kind; and men would cooperate more if their opponents are women). In reality, however, these knowledge are not necessarily conscious to the participants. Therefore, the set of experiments could be additionally separated into two conditions, one where the participants are given a fact sheet of these gender-related cooperation preferences, and one where they aren't. It would be quite interesting to understand this effect.

**We did not assume in our experiment – rather, we hypothesised based on previous studies – that gender pairing would have some implications on how men and women play the strategic game. But we do take your point and agree that it would be interesting to explore in the future the implications of making the gender effects more salient before the game. This is, however, beyond the scope of the current study, which we have pre-registered our hypotheses prior to conducting the experiment. 

In summary, the referee would like to see an improvement of the method description and additional experimental results that considered the effect of the awareness of gender preferences, and the opponent's experimental settings. Other than these, the research can illuminate important insights on critical real-world interactions in our digital life.

**Thank you very much for all these valuable comments.

—

Reviewer #3: Summary

The paper entitled Large Losses from Little Lies: Strategic Gender Misrepresentation and Cooperation investigates an interesting question on cooperation with the possibility of misrepresenting one’s gender. The lab and the field experiment study provide large-sample and multinational evidence that confirm the harmful impact of the available deception on gender. This paper extends mixed previews findings on gender differences in cooperation and consider gender stereotype. Overall, this paper is interesting and helpful in understanding the relationship between small lies on gender and cooperation. However, I have some concerns which may help during the revision.

Major Comments

1. What are the purpose and settings of the fourth treatment- Randomly assigned gender? Does this treatment involve deception on subjects? Even if not, I think this design is counterfactual and unreasonable. It is almost impossible to force people to misrepresent their gender in reality. So, not surprisingly, the result of the treatment is disappointing. By contrast, the treatments in regard of hidden gender or gender composition of populations make more sense.

**We have stated on p.13 that the fourth treatment was setup to test the importance of agency, i.e., the freedom to choose whether to misrepresent. No, it does not involve deception because we were explicitly clear to both participants in a pair that one person will be randomised a gender that either the same or different from their true gender. We have made this point clearer in the experimental method section; see p.13. But in saying that, we do agree with you as to why the result of this treatment is disappointing. Nevertheless, it did give us some insight into the importance of agency in this context.

2. There might be some exciting findings in the verbal and anonymous communication before the game. Did the participants have the chance to send gender messages in this part? If so, these messages might be part of the effect of gender misrepresentation or bias in the estimation. Secondly, since this communication is unstructured and participants can talk about whatever they want, does the text data reflects some interesting behavior pattern? For instance, did the participants misrepresenting gender grab this chance to convince their partner to believe them? The authors should consider digging deep into the text data.

**Thank you very much for this suggestion. We followed your first suggestion and conducted a text analysis by treatment (see Fig.1A in the online appendix) and found no differences in the messages being communicated across treatments. We, however, did not conduct a specific text analysis on those who misrepresented their gender, partly because of the much smaller sample size. Also, it is partly because, as we have shown in the paired decision-making regressions, they were not very successful in convincing the other partner to cooperate whilst they choose to split. 

3. The consistency between lying behaviors and stealing/splitting behavior should be checked. Some participants might lie to take advantage of stereotypes. However, there might be some participants who were afraid of defection. Once they get the chance to misrepresent, they might lie about their gender to protect themselves from others' defection. This type of participant might be more willing to cooperate. The authors should analyze the different behavior patterns further.

**We did this in Table 6 and find that women who misrepresented themselves as men in the mixed-sex pair are less likely to defect. However, the coefficient is statistically insignificantly different from zero. But because the sample size is very small, we cannot definitively conclude that the insignificant result is not due to having low power. By contrast, the strongly significant effect of women misrepresenting themselves as men in the same-sex pair cannot be the outcome of low power.

4. One of the most interesting results is the different strategies used by male and female. Which strategy is closer to the rational expectation or which is the more successful strategy? Is there any deeper psychological explanation for this result?

**Since Table 5’s results suggest that people who misrepresented are likely to defect, we’re not sure whether the strategy that follows the decision to misrepresent is a successful strategy. This is largely because people who knew that they may or may not be playing against someone who is mispresenting their gender are also less likely to cooperate. We are also not sure there are deeper explanations to the results as we did not collect data on beliefs. For that, we do apologise that we are unable to address this point as fully as we would like. 

5. Last but not least, the regression coefficients of most treatments are not significant, which means the impacts are small statistically. To make conclusions more convincing, maybe more explanations are needed. What’s more, the analysis on mechanism is somewhat too simple and superficial, it should be more in-depth, for example, including people’s beliefs which are crucial to the results.

**Thank you very much for this suggestion. Following Referee 1’s suggestions, the latest results on the treatment effects are positive and statistically robust. Unfortunately, we did not include questions about people’s beliefs in the experiment. We regret not having done so, and as a result have decided to put this as a shortcoming of the experiment in the conclusion.

Minor comments

1. There are several mistakes in the paper. In Introduction section, (p4 line2) "a variant" repeated for 2 times. In Empirical strategy section, p14, paragraph 4, line 3 and 4, letters of the equation failed to be shown, perhaps due to the document export problems.

**Done.

2. I suggest adding some non-parametric statistical hypothesis tests before skipping into a probit regression, which may help explore differences between two treatments and focus on the significant outcomes. Figure 2 alone is insufficient to clarify the overall treatment effect of four conditions.

**Done. See the Kruskal-Wallis equality-of-population test below.

3. Wilcoxon Signed Rank in the second paragraph on Page 15 is misused. As mentioned above, the Wilcoxon Rank Sum Test is the test that is used in a between-subject design. It should be checked

**Thank you for this suggestion. Considering this, we have decided to replace the Wilcox Signed Rank test for another nonparametric test, i.e., the Kruskal-Wallis equality-of-population test.

4. The reference list are not completed. For example, “Kim (2020)” in the page 14 is not included in the references.

**Done.

---

## [Decision Letter · Decision Letter 1]

14 Feb 2023

Large Losses from Little Lies: Strategic Gender Misrepresentation and Cooperation

PONE-D-22-25168R1

Dear Dr. Powdthavee,

We’re pleased to inform you that your manuscript has been judged scientifically suitable for publication and will be formally accepted for publication once it meets all outstanding technical requirements.

Kind regards,

Luo-Luo Jiang, Ph.D.

Academic Editor

PLOS ONE

**Comments to the Author**

1. If the authors have adequately addressed your comments raised in a previous round of review and you feel that this manuscript is now acceptable for publication, you may indicate that here to bypass the “Comments to the Author” section, enter your conflict of interest statement in the “Confidential to Editor” section, and submit your "Accept" recommendation.

Reviewer #1: All comments have been addressed

Reviewer #2: All comments have been addressed

Reviewer #4: (No Response)

2. Is the manuscript technically sound, and do the data support the conclusions?

Reviewer #1: Yes

Reviewer #2: Yes

Reviewer #4: Yes

3. Has the statistical analysis been performed appropriately and rigorously? 

Reviewer #1: Yes

Reviewer #2: Yes

Reviewer #4: Yes

4. Have the authors made all data underlying the findings in their manuscript fully available?

Reviewer #1: Yes

Reviewer #2: No

Reviewer #4: Yes

5. Is the manuscript presented in an intelligible fashion and written in standard English?

Reviewer #1: Yes

Reviewer #2: Yes

Reviewer #4: Yes

6. Review Comments to the Author

Reviewer #1: The authors have sloved most of my concerns except my comment on literature. The mispresenting of gender is a kind of cheating\\lying behavior. So I believe some literature on lying should be related to this study.

Reviewer #2: The referee would like to thank the authors for the revision and response to the reviews. The revision and response address most of my concerns.

Reviewer #4: This work discusses an interesting question regarding gender differences in human cooperation and explores its extended social preference through empirical text. The kernel of the experiments is the gender misrepresentation of self-identity in a dilemma game, referring to a game-theoretic system, and the mispresented information is based on intended action. By looking through the revised version of the manuscript, I think the authors have well addressed the critical issues raised by other reviewers. The experimental assumptions and methods adopted are reasonable and reliable, and the conclusion is robust. However, I’ve noticed several points that make me draw the conclusion that the MS with the current style cannot be interim accepted, despite I embrace with a positive feeling.

1. In the Background and hypotheses Section, the author mentioned one leading mechanism named indirect reciprocity, which can promote the evolution of cooperation among human society. The author states that the establishment of indirect reciprocity is based on the interactive pair or group interaction. However, is this also related to the Golden Ball game used in their experiment?

2. Also, in B&H Section, the authors specifically mentioned indirect reciprocity and punishment mechanism. So, this kind of costly punishment can be denoted as direct reciprocity. If the author wants to briefly introduce the potential mechanisms to promote cooperation, they may refer to some classic papers, like Science (314) pp. 1560-1563. (It does not mean you have to cite this paper.)

3. The authors argue that gender differences are one of social preference. Many papers have been released discussing the influence of social preference on the evolution of cooperation during the past decades. They may explain this in more detail.

4. I noticed that some of their math formulas have boxes (page 15), and some do not (page 22). Please make sure the formulas are consistent.

7. PLOS authors have the option to publish the peer review history of their article (what does this mean?). If published, this will include your full peer review and any attached files.

Reviewer #1: **Yes: **Shuguang Jiang

Reviewer #2: **Yes: **Baihan Lin

Reviewer #4: No

---

## [Editor Report · Acceptance letter]

21 Feb 2023

PONE-D-22-25168R1 

Large Losses from Little Lies: Strategic Gender Misrepresentation and Cooperation 

Dear Dr. Powdthavee:

I'm pleased to inform you that your manuscript has been deemed suitable for publication in PLOS ONE. Congratulations! Your manuscript is now with our production department. 

Kind regards, 

on behalf of

Dr. Luo-Luo Jiang 

Academic Editor

PLOS ONE